# Exclusion from Social Relations in Later Life: Micro- and Macro-Level Patterns and Correlations in a European Perspective

**DOI:** 10.3390/ijerph182312418

**Published:** 2021-11-25

**Authors:** Thomas Hansen, Marcela Petrová Kafková, Ruth Katz, Ariela Lowenstein, Sigal Naim, George Pavlidis, Feliciano Villar, Kieran Walsh, Marja Aartsen

**Affiliations:** 1Department of Mental health and Suicide, Norwegian Institute of Public Health, 0213 Oslo, Norway; 2NOVA Norwegian Social Research, Oslo Metropolitan University, 0130 Oslo, Norway; maraar@oslomet.no; 3Department of Sociology, Faculty of Social Studies, Masaryk University, Jostova 10, 60200 Brno, Czech Republic; kfkv@seznam.cz; 4Max Stern Yezreel Academic College, Haifa 31905, Israel; ruth@soc.haifa.ac.il (R.K.); sigal.naim@gmail.com (S.N.); 5Center for Research & Study of Aging, The University of Haifa, Haifa 31905, Israel; ariela@research.haifa.ac.il; 6Department of Culture and Society, Linkoping University, 60230 Norrkoping, Sweden; george.pavlidis@liu.se; 7Department of Cognition, Development and Educational Psychology, University of Barcelona, 171, 08035 Barcelona, Spain; fvillar@ub.edu; 8Irish Centre for Social Gerontology, National University of Ireland Galway, H91 F677 Galway, Ireland; kieran.walsh@nuigalway.ie

**Keywords:** social exclusion, social relationships, older adults, Europe, SHARE data, gender

## Abstract

Older adults face particular risks of exclusion from social relationships (ESR) and are especially vulnerable to its consequences. However, research so far has been limited to specific dimensions, countries, and time points. In this paper, we examine the prevalence and micro- and macro-level predictors of ESR among older adults (60+) using two waves of data obtained four years apart across 14 European countries in the Survey of Health, Ageing and Retirement in Europe (SHARE). We consider four ESR indicators (household composition, social networks, social opportunities, and loneliness) and link them to micro-level (age, gender, socioeconomic factors, health, and family responsibilities) and national macro-level factors (social expenditures, unmet health needs, individualism, social trust, and institutional trust). Findings reveal a northwest to southeast gradient, with the lowest rates of ESR in the stronger welfare states of Northwest Europe. The high rates of ESR in the southeast are especially pronounced among women. Predictably, higher age and fewer personal resources (socioeconomic factors and health) increase the risk of all ESR dimensions for both genders. Macro-level factors show significant associations with ESR beyond the effect of micro-level factors, suggesting that national policies and cultural and structural characteristics may play a role in fostering sociability and connectivity and, thus, reduce the risk of ESR in later life.

## 1. Introduction

Combatting old-age social exclusion is a major target in European policy [1]. Social exclusion is multidimensional, covering multiple domains of life, and a central component relates to exclusion from social relations (ESR). ESR is multifaceted in nature and can be understood as a situation of being socially and emotionally disconnected from social relations, with both objective and subjective components to its construction and manifestation [2]. While objective ESR can be described as having a limited social network, infrequent social interaction, or lack of participation in social activity, subjective ESR can be characterized by a perceived lack of companionship or feelings of loneliness [2]. These are overlapping yet distinct dimensions, as older men and women may be socially isolated but not feel lonely, and vice versa. Being socially connected is a universal basic human need [3], and not being connected has manifold consequences for the person as well as for society as a whole [4]. Both objective and subjective components of ESR have been consistently and quite equally associated with physical and mental health and mortality for both genders [5,6,7]. Therefore, identifying key risk factors that predict objective and subjective outcomes of ESR across all its dimensions is critical for understanding and preventing this form of disadvantage and guiding the mode and timing of possible interventions for older men and women.

However, compared to the burgeoning literature on material and financial exclusion, especially among children and young adults, little attention has been paid to comprehensive forms of social disadvantage, such as ESR, for older age groups. This deficit is surprising for at least two reasons. First, ESR is closely aligned with the policy aim to combat older age isolation and loneliness evident in many European states. Second, for older adults, ESR is possibly more relevant than other measures of deprivation: it is the form of exclusion that has been shown to increase the most in later life [8], with consequences for people’s capacity to maintain independence and well-being, and to overcome other forms of disadvantage in older age [9]. Increased prevalence among older men and women may also reflect an accumulation of contributing factors across the life course, combined with fewer opportunities to build new relational ties [1].

Within the extant literature on ESR, three critical gaps can be identified. First, few studies have adopted a multidimensional approach to exclusion in the social domain, but instead, analyzed either distinct dimensions of ESR (such as loneliness) or a composite index score [2,10]. Studies suggest that among older adults, age is negatively related to some dimensions of social connectedness (e.g., network size and living with a partner), but positively related to others (socializing with neighbors and volunteering) [2,10]. A multidimensional approach promotes holistic understandings precluded in the analysis of one or two sub-dimensions, and an absence of such an approach fundamentally reduces the capacity to identify the multiple modes and timing of necessary interventions. Furthermore, although it promotes parsimony, a disadvantage of an index is that scores—other than the two extremes (high or low on all components)—are hard to interpret and obscure information that might be relevant such as how different dimensions may relate to each other. ESR is a wide-ranging and complex phenomenon and is therefore not easy to represent with a single figure [1]. Additionally, from a pragmatic perspective, there are potential pitfalls in developing rules for aggregation and weighting of data and problems with variations in the quality and availability of appropriate data [11,12]. The complexity of relational dimensions, and of the processes and experiences that influence these dimensions, highlights the need for more comprehensive forms of analysis of social deprivation.

Second, few studies interrogate the role of gender in multifaceted forms of ESR in later life. Doing so is of fundamental importance as the different life-course trajectories of men and women may lead to different risks for the different dimensions of ESR in old age. More specifically, ESR may be unequally high for older women as they score higher than older men on risk factors such as financial distress, health problems, widowhood, and family caregiving [13]. Findings on gender inequalities in ESR are somewhat mixed, however. While some studies show older men to be more often excluded from social relationships than women [10,14], other studies show women to be more vulnerable to loneliness [15]. These patterns may be especially acute in Eastern Europe, where women face pronounced risks of ESR as relatively many of them are aging without a partner and with health problems and financial concerns [16]. Not surprisingly, therefore, in the former socialist countries but not in Northwestern European countries, rates of loneliness among older adults have been found to be 5–15 percent higher for women than for men [15]. However, an opposite pattern is evident for contact frequency with family and relatives, and a mixed pattern exists for contact frequency with friends and access to instrumental and emotional support [13].

Third, there are few cross-national comparative studies and few explorations of the role of macro-level factors. These shortcomings prevent conclusions on contextual drivers of ESR and hamper our ability to inform the debate about public health policy relevant to older age groups. Given the striking national differences in the living conditions of older adults and in the degree to which cultural and institutional frameworks provide opportunities for connectivity and social participation, we may expect major country-level differences in ESR to exist [2]. For instance, more generous welfare systems may more readily allow persons to demonstrate greater competence and mastery over their lives, facilitating greater social participation and integration, if desired, and promoting healthy psychological states that reduce the risk of late-life ESR [17]. High social spending could thus be assumed to protect against some of the risks of ESR, for example, by providing better health care and social services, income and housing conditions, public transport, and support to family caregivers [18]. Another potentially relevant country-level factor is trust. Trust forms the foundation of a well-functioning society, and can be defined as either institutional trust towards public institutions or interpersonal social trust towards other people [19,20]. Social trust has been posited to play a crucial role in the formation and maintenance of interpersonal relationships [21,22]. As Svendsen [23] explains, people who learn to trust others will interpret their expressions and gestures as less threatening, and will be able to relate more immediately to them, and therefore improve the conditions necessary for forming attachments. Likewise, institutional trust is critical for the smooth operation of all interactions between governmental institutions and citizens [24]. Low levels of institutional trust can compromise social connectivity by eroding social trust and by its association with social unrest, crime, and lower social solidarity [24]. These notions, coupled with the strong northwest to southeast divide in social and political trust in Europe [25,26], predict a corresponding divide for social connectivity. Furthermore, some nation states may culturally, or even politically, emphasize notions of solidarity and relational practices that are more typically associated with cohesion and socialization and, thus, effectively both bolster social resources and alter relational expectations. For example, Southern and Eastern European countries are generally characterized by a more familistic and collectivistic orientation than the individualism associated with Western countries [27,28]. Individualism can be defined as a preference for a loosely knit social framework in which individuals are expected to take care of only themselves and their immediate families [29]. Its opposite, collectivism, represents a preference for a tightly knit framework in society in which individuals can expect their relatives or members of a particular in-group to look after them in exchange for unquestioning loyalty. Because they tend to value and expect strong ties within the family and community, familistic cultures may prevent aspects of ESR by promoting social integration. Research suggests, however, that some dimensions of ESR, such as those regarding loneliness, social participation, and social networks, are more pervasive in southern and eastern states than in northern and western European nations, suggesting complex interactions between different sets of macro-variables [13,15,30]. However, although people in individualistic countries tend to be less lonely due to favorable health and financial conditions, little is known about whether an individualistic culture (an orientation towards independence and personal rewards) in itself is associated with ESR [15].

This paper addresses the aforementioned gaps by examining the nature and predictors of ESR among older men and women (60+) using two waves of data across 14 European countries. We consider four indicators of ESR (household composition, social network, social opportunities, and loneliness) and link them to micro-level (age, gender, socioeconomic factors, and health) and macro-level factors (social expenditures, unmet health needs, individualism, and trust). More specifically, we ask: What is the prevalence of ESR in European countries for older men and women, and which micro- and macro-level factors contribute to explain variation in ESR?

## 2. Materials and Methods

### 2.1. Data

We use data from the fourth (2011) and sixth wave (2015) of the Survey of Health, Ageing and Retirement in Europe (SHARE). SHARE is a multidisciplinary and cross-national household panel survey that collects detailed information on socioeconomic status, health, and social and family networks for nationally representative samples of community-dwelling individuals aged 50+ [31]. We include 14 countries that were included in both waves, representing Northern (Denmark, Sweden), Central (Austria, Belgium, France, Germany, Switzerland), Southern (Italy, Portugal, Spain), and Eastern (Czech Republic, Estonia, Poland, Slovenia) Europe. Respondents with missing observations in the independent variables are excluded in the analytical structural equation model. We use data from 9795 women and 6558 men aged 60+ in 2011. We focus on those aged 60 and above because the age of 60 is widely considered as the onset of old age [32].

### 2.2. Dependent Variables

ESR is measured with four indicators: household composition, social network, social opportunities, and loneliness. The social network indicator will be examined by network typologies. These indicators cover not only the breath of our conceptualization of ESR, but also the three “spheres of sociability” [33]: the primary sphere, secondary sphere, and tertiary sphere. The primary sphere of sociability shows the household structure of individuals, i.e., whether or not a person lives with others in the household. The secondary and tertiary sociability spheres depend on informal and formal participation of individuals in society, i.e., whether or not a person meets friends or relatives, and whether or not a person socially participates in organization(s) or relevant activities.

Household composition has three categories: lives with partner, not with children; lives with partner and children; and lives with no partner or children. We exclude the few (N = 51) who live with children but no partner.

Social network, or the collection of interpersonal ties [34], is defined as a nominal variable characterizing the social network type (see below), based on six indicators (network size, proximity, contact frequency, network satisfaction, perceived closeness, and proportion of the network that is a family member). As the partner is already included in the household composition variable, it is not included in the social network. The identification of network members is based on a name-generating inventory, introduced in the fourth round of SHARE, in which respondents were asked to personally identify up to seven persons with whom they discussed important matters [35]. Network size reflects the number of network members identified (range 0–7) with whom the respondent has discussed important things in the last 12 months. Proximity is the number of people counted under network size as network member living within a radius of 25 km (range 0–7). Contact frequency is the average contact frequency with the network members (0 = no network members, 1 = never, 2 = less than once a month, 3 = about once a month, 4 = about every two weeks, 5 = about once a week, 6 = several times a week, 7 = daily). Satisfaction reflects how satisfied people are with their social network, on a scale from 0 (completely dissatisfied) to 10 (completely satisfied). Felt closeness, the average perceived closeness with the network members, is assessed by asking how close the respondent feel to each network member, from “not very close” (1) to “extremely close” (4). Proportion of family members is the number of non-resident family members (e.g., parent, grandchild, and niece) divided by the network size. Due to high skewness, this variable was categorized into 0 (no family members), 1 (fewer than half of the network are family members), and 2 (more than half of the network are family members).

Social participation reflects the number of activities performed weekly or more often. Activities include volunteering/charity work, caring for sick/disabled adult, helping family/friends/neighbors, attending an educational/training course, taking part in a sport/social/other club, taking part in a religious organization, and taking part in a political/community/religious organization.

Loneliness is measured by a combination of a single item and a short version of the Revised-University of California at Los Angeles (R-UCLA) scale [36,37]. The scale was based on three questions: How much of the time do you … (1) feel a lack of companionship, (2) feel left out, and (3) feel isolated from others. The single item asks, “Do you feel lonely?” All four items have the responses options (1) often, (2) some of the time, and (3) hardly ever or never. The combined scale ranges from 4 to 12 (high loneliness) (α = 0.82). We also dichotomized loneliness into not lonely (score < 7) and lonely (score ≥ 7).

### 2.3. Independent Variables

We include in our models the following *micro-level* factors potentially associated with ESR [2]. Gender (1 = men, 2 = women) and age (in 2011) are included, as isolation and loneliness are generally associated with female gender and older age. We include two socioeconomic variables and one health measure as such personal resources can facilitate social participation and foster supportive network ties and access to support [2]. Educational level is classified based on the country-specific ISCED-97 codes [38] into low (level 0–1), medium (2–4), and high (5–6). Financial situation (perceived difficulties in making ends meet) ranges from 1 (great difficulties) to 4 (very comfortable). Health limitation is measured with the Global Activity Limitations Index (GALI) [39,40], which asks: “In the past 6 months, to what extent have you been limited because of a health problem in activities people usually do?” Response categories are severely limited (1), limited, but not severely (2), and not limited (3). Family responsibilities reflects whether “Family responsibilities prevent me from doing what I want to do”, from often (1) to never (4). Such responsibilities can entail taking care of an ailing partner of parents, and can compromise social engagement and participation [41]. We chose this measure over a direct measure of caregiving (no/yes) as it gauges not only care but also whether it compromises other engagements.

We include five potential macro-level determinants of ESR that reflect commonly used indicators of cultural, socioeconomic, and welfare conditions. These variables are merged manually with the SHARE data. Information about social expenditure was derived from the online database Eurostat (https://appsso.eurostat.ec.europa.eu/nui/show.do?dataset=spr_exp_sum&lang=en; accessed on 13 August 2020) and covers a country’s total expenditures (in 1000 EUR per inhabitant) for sickness/health care, disability, old age, family/children, unemployment, housing, and social exclusion (https://ec.europa.eu/eurostat; accessed on 13 August 2020). Information about unmet health care needs is also derived from Eurostat and reflects the percentage of the population who agree with the statement that there was a time in the previous 12 months when they felt they needed health care or dental care but did not receive it. Data on individualism are derived from Hofstede’s (2011) model of six dimensions of national cultures (www.hofstede-insights.com/product/compare-countries; accessed on 11 July 2020). The individualism–collectivism continuum ranges from 0 (collectivistic) to 100 (individualistic). Data on social and institutional trust are the aggregated scores at the country level derived from the European Social Survey (ESS), wave 6 (year 2012) (www.europeansocialsurvey.org; accessed on 11 July 2020). The ESS is a biennial cross-national survey that has been conducted across Europe since 2001 (https://www.Europeansocialsurvey.org/about/; accessed on 11 July 2020). In line with the recommendations of the European Social Survey, we weighted the dataset when calculating the mean scores by using the post-stratification weight in combination with the population size weights. *Social trust* reflects the mean score on three 0–10 questions: “Generally speaking, would you say that most people can be trusted, or that you can’t be too careful in dealing with people?”, “Do you think that most people would try to take advantage of you if they got the chance, or would they try to be fair?”, and “Would you say that most of the time people try to be helpful or that they are mostly looking out for themselves?” (α = 0.73). Institutional trust reflects mean scores to three 0–10 questions: “How much you personally trust each of the following institutions: politicians, the country’s parliament, and the police” (α = 0.76). For each country, the average of the individual means was used as a macro-level variable in the analyses. A higher score on both trust scales indicates a higher level of trust.

### 2.4. Statistical Analyses

In a first step, we estimated a social network typology by means of a Latent Class Analysis [42]. The decision on the number of network types was based on a combination of principles, including parsimony, class sizes not smaller than 5 percent, and high (>0.80) entropy, which is indicative for classification quality. The four-class solution provided the best fit to the data. Although the five-class solution had a lower BIC than the four-class solution (BIC = 9985895 and 1013049, respectively) and higher entropy (0.90 and 0.87, respectively), we preferred the four-class solution as adding a fifth class resulted in a less parsimonious description of the data (the Vuong–Lo–Mendel–Rubin likelihood ratio test for four versus five classes was not significant (*p* = 0.74)). The description of the network types, or classes, follows from the conditional probabilities (see Table 1). These can be interpreted as follows. Type I people can be considered at risk of being excluded from a supportive social network. These people are characterized as having a very small network (0.6 persons on average). The contact frequency is low, although they feel extremely close to the network member(s), who tend to be non-family. Type II people have a small network (2–3 people), which they contact several times per week to daily. Type III people have a medium-sized network (3–4 people), comprising family members and other people, half of whom live in close proximity. Type IV people have the largest network (5–6 people), mainly family members, who mostly live in close proximity. As indicated by the conditional probabilities (Table 1, row 2 to 9), the four network types differ from each other with respect to network size, proximity, contact frequency, and proportion of family members. Network satisfaction and closeness do not discriminate across the four types, as all conditional probabilities are equally high for the four network types (satisfaction is close to 9 on a scale from 0 to 10 for all four types and all feel at least very closely connected to the members of the network). The latent class probabilities (first row) suggest that the prevalence of people with network type I, II, III and IV is 29, 42, 17, and 12 percent, respectively. The most likely class membership was saved and merged with the data file for further analysis.

Second, we described levels of ESR for each dimension and each country, separately for men and women. Third, we estimated a structural equation model (SEM) with the four dependent variables and the selected micro- and macro-level independent variables (Figure 1). Each dependent variable, measured at wave 6 (2015), was regressed on micro-level variables measured at wave 4 (2011), and macro-level variables for 2011 (or an adjacent year if information for 2011 was unavailable). The SEM consists of a series of linear regressions for the continuous outcomes (social participation and loneliness) and a series of multinomial regressions for the nominal outcomes (intimate relations and social network). The advantage of SEM over traditional regression techniques is that it can take into account potential high correlations among outcomes, measurement error, and missing observations. The model is estimated with Mplus version 8.4 [43]. The model is estimated using Monte Carlo integration with 500 integration points. We further use the Robust Maximum Likelihood (MLR) estimator that is available in Mplus to account for missing values. MLR is considered superior to the Maximum Likelihood estimation when data are categorical and when assumptions of normality are violated [44].

## 3. Results

Table 2 shows the bivariate correlations among the variables of this study, separately for men and women. Given the large sample size, virtually all correlations are significant at the 0.05 level. As shown, intercorrelations among the independent variables are low-moderate, suggesting that multicollinearity is not a major concern.

Table 3 shows the prevalence of ESR across gender and European regions and countries. As shown, the rates of people who live without a partner are highest in the eastern countries and lowest in the southern countries, and generally higher among women than men. Type I (very limited) networks, however, are more common among men, and the rates are higher in the southeast than in the north–central European countries. Low social participation and high loneliness are more common in the southern and eastern countries, and while there are no gender differences for social participation, loneliness is generally more often reported by women.

Table 4 shows the results of the structural equation modeling of indicators of ESR for men and women. Among men, higher age, lower education, health limitations, and financial problems are all associated with most but not all indicators of ESR. One exception is that Type I social network is only associated with education. Family caregiving is related to higher loneliness and, as expected, living with a partner.

Similarly for women, higher age, lower education, health limitations, and financial problems generally but not consistently relate to higher likelihood of ESR. One interesting exception is that living with no partner or children is associated with high education, i.e., the opposite of what emerged for men. Caregiving is associated with higher loneliness, but lower ESR in terms of living situation and social participation.

At the macro level, higher social expenditure predicts living with a partner, higher social participation, and less loneliness (men only). Higher levels of unmet health needs relate to living with a partner and/or children, having a smaller social network, and more loneliness (women only). Higher individualism and higher trust predict more social participation and lower loneliness, but have inconsistent associations to the other ESR indicators.

## 4. Discussion

The main goal of the study was to analyze and describe the prevalence and micro/macro predictors of ESR among men and women, aged 60 years and over. We use SHARE data from 14 countries and analyze four different indicators of ESR (household composition, social network size, social participation, and loneliness). Findings reveal, with some qualifications, a northwest to southeast gradient, with the lowest rates of ESR in the stronger welfare states of Northwest Europe. A general pattern is that the Nordic countries are characterized by the lowest and the Eastern European countries by the highest level of ESR. These patterns echo those observed in prior studies on social network size [45] and loneliness [15,46]. Much of the country variation is attributable to compositional differences, as notable risk factors of ESR among older adults (e.g., fewer socioeconomic and health resources) are the most prevalent in the southeast. In addition, macro-level factors show significant associations with ESR beyond the effect of micro-level factors, suggesting that national policies and broader cultural and structural patterns play a role in fostering social inclusion and cohesion. Of note, a higher level of individualism and greater trust in other people predict higher social participation and lower levels of loneliness. The findings support the often-alleged role of interpersonal trust in fostering social connectedness and reducing loneliness [26].

Our findings align with prior research suggesting that generous and comprehensive welfare supports facilitate and encourage social integration and social participation [17]. While, in the past, old-age social welfare systems were primarily focused on providing support and care, they are now increasingly focused on “active ageing” and a more “participation-centered” service. These focuses emphasize, for example, a healthy lifestyle, community and civic engagement, employment, active leisure, and social relationships, which in turn contribute to reduce and prevent old-age social exclusion [47]. The high social participation in Northern Europe may also reflect that this region has higher organized (formal) participation, whereas Southern Europe may have more informal participation. For example, in the Northern European countries, formal voluntarism is more normative and prevalent, and institutionally encouraged and supported [48]. The unfavorable position of Eastern European older adults in terms of their social relationships, social participation, and loneliness levels reflect the comparatively high rates of poverty, health problems, and bereavement among—especially—women in this region [15,49]. To date, scant attention has been paid to the risk of late-life ESR in Eastern Europe, some of the European countries with the most severe challenges in caring for the material, social, and health needs of their older populations [18,50].

The loneliness divide can also reflect that Southern Europeans may have higher expectations, and there may be differences in the degree of what constitutes intimacy and closeness. Several authors point to the importance of considering people’s frames of reference and normative orientations in the cultural context of countries under investigation [15]. Loneliness occurs when the quality of one’s social relationships falls short of the expected or desired quality of social relationships. Johnson and Mullins [51] introduced the term ‘‘loneliness threshold’’ to refer to the level at which loneliness arises. Southern and Eastern Europeans, because of high expectations of strong family and community ties, may have a lower loneliness threshold than other Europeans. A low loneliness threshold may make matters worse for older adults in countries with high rates of widowhood, decreasing fertility rates, and increasing outmigration [15]. Moreover, it may be that political upheavals, economic insecurity, and greater socioeconomic inequalities have eroded feelings of trust and social integration, which in turn increased the risk of loneliness among older adults in Eastern Europe [18,52]. In sum, the combination of a low loneliness threshold and negative changes in social integration may help to explain high levels of loneliness in former socialist countries.

We also find that women, compared with men, are more at risk of ESR as measured by living alone and loneliness. Women’s relative vulnerability corresponds with prior studies on social isolation [2] and loneliness [15], and seems largely attributable to their greater likelihood of widowhood. Women are generally not more socially excluded as measured by interpersonal ties or social participation, however. This finding challenges previous studies showing that older women tend to report significantly larger social networks than their male counterparts [53], but dovetails with studies showing that gender differences in social interaction and social network size diminish or disappear in late adulthood [54]. While women’s higher risk of loneliness is likely mainly driven by gender differences in living situation, more “subjective” factors could also play some role. Insofar as women generally are more socially active and integrated, at least in early and middle adulthood [54], their potentially higher social standards and expectations may cause vulnerability to loneliness when experiencing social losses in later life [15,55]. Findings also underscore how gendered risks vary across countries characterized by different preconditions for social contact and participation. We find relatively few social relationships, low social participation, and high levels of loneliness among Eastern European aged people, especially among women. This may be due to high rates of poverty and health problems [15,49] that, in turn, increase susceptibility to ESR in later life.

A particularly interesting finding is the seeming disconnect between objective ESR and feelings of satisfaction with the social network: even those with a very limited social network (“Type 1”) report, on average, score 9 (out of 10) on network satisfaction. This illustrates the notion that social isolation does not equate with dissatisfaction and unmet social needs. This phenomenon, labelled “positive solitude” (PS), is found to be a volitional and enjoyable experience for many older adults [56]. Importantly, there is often a gap between old people’s experience of PS and the perceptions of health professionals regarding this experience. This is an important issue as older people, especially those in a long-term care facilities, may need PS to gain a sense of control and freedom as part of their quality of life [56]. Hence, even when studying ESR, one must remember that despite many negative implications, there may also be some positive benefits.

## 5. Conclusions

This study has several limitations that suggest avenues for future research. One weakness is our inability to reveal causal order given the lack of longitudinal (within-person) analysis. Thus, it remains unclear whether the correlates—e.g., low socioeconomic status, being a caregiver, poor physical or mental health, and disability—are drivers or outcomes of ESR. Concomitantly, future longitudinal analysis should adopt a life-course perspective and assess the impact of earlier events and conditions on later-life ESR—for example, the impact of life events (e.g., widowhood, retirement, and moving residence) and of critical risk junctures for the development of ESR.

Despite these limitations, and as was its central aim, this study makes a number of contributions to international scholarship on ESR, and advancing understanding across a number of dimensions. First, the findings, and the focus on an extended range of ESR indicators, underscore the need to expand research on relational deprivation beyond loneliness and isolation. As illustrated here, committing to a more comprehensive and multidimensional approach to studying ESR can provide a more nuanced and insightful understanding of the manifestation of this phenomenon. Second, our analysis provides further clarity in relation to questions around gendered experiences of certain aspects of ESR. In doing so, this paper adds to the weight of evidence that indicates older women’s enhanced vulnerability to isolation and loneliness. This is in addition to confirming that further work is required to disentangle the complex interactions between gendered and cultural norms across countries in terms of social contact and participation. Third, the scope of the multi-level independent variables included in this analysis demonstrates the nature and value of assessing broader structural and upstream causes of exclusionary processes and outcomes, with a view to their mitigation. The European cross-national focus illustrates the worth of adopting a cross-national perspective to unpack the relative social contextual elements of ESR, and understand the structural causes at play. It also illustrates how national policies and cultural and structural characteristics may serve to foster sociability and connectivity, and thus reduce the risk of ESR in older age. Ultimately, it is only by adopting comprehensive analysis approaches, that are gender and context sensitive, and that incorporate objective and subjective dimensions, we can meaningfully progress scientific understanding of ESR in later life, and hope to reduce its prevalence and impact across Europe.

Our findings somewhat predictably reaffirm the need for person-centered approaches in combatting complex forms of disadvantage, such as exclusion from social relations [57]. While this requires, and can be facilitated, by the inclusion of subjective and objective measures in assessing such potential deficits, it is best executed in working with older people directly, and their existing networks to identify the sorts of social needs, preferences and priorities that allow for that targeting of specific interventions. Despite some of the associations identified in this research, not everyone living alone or with fewer ties may judge themselves to be excluded, and for all intents and purposes may not be. A person-centered focus will help account for such preferential and circumstantial elements [57]. However, the findings also suggest that there may be interlocking sets of conditions that may increase the breadth and depth of exclusion from social relations, such as sex and gender, social inequality, more loneliness and less participation. A focus on multiple exclusions, and in some cases, the more active consideration of the life events and transitions that have created them, is likely to be particularly valuable.

In local and municipality settings, each of these considerations might be usefully delivered through existing age-friendly programmes, given their prevalence across European contacts, higher integrity, and their integrative and multiagency approaches to support older people [57,58]. Targeting exclusion from social relations implicates more than just interventions focused on building connections and relationships, and instead, draws in the sort of outdoor spaces and amenity infrastructure that can facilitate social outlets and social contact—the sort of transport that, through its accessibility, frequency and chosen destination, enable great connectivity; and the sort of services targeting other needs that, by their nature of delivery, connect people in community settings (e.g., health clinics) and that, as a by-product of their design, monitor and broker other supports for those who increase adults’ social connectivity (e.g., postal services).

## Figures and Tables

**Figure 1 ijerph-18-12418-f001:**
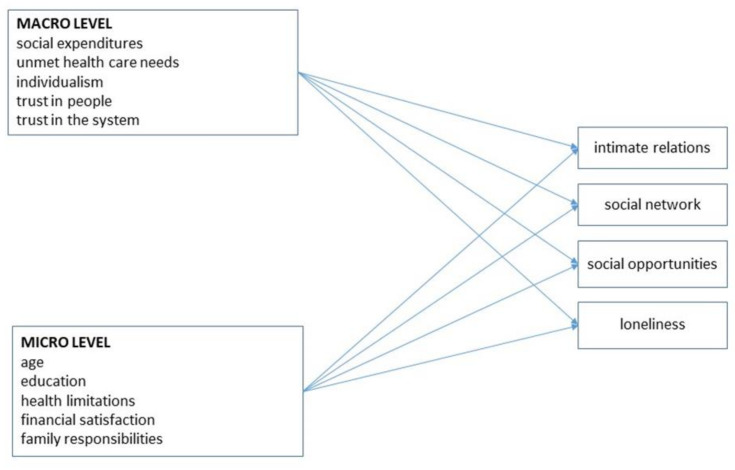
Theoretical model for relationships between micro- and macro-level variables (wave 4) and four indicators of ESR (wave 6).

**Table 1 ijerph-18-12418-t001:** Latent class probabilities and conditional probabilities for four network types.

Class	I	II	III	IV
Latent class probabilities	0.29	0.42	0.17	0.12
Network size (0–7)	0.56	2.59	3.69	5.63
Proximity	0.01	0.85	1.16	3.45
Contact frequency	0.03	6.19	4.62	5.26
Satisfaction (0–10)	8.95	9.17	8.61	9.03
Felt closeness (ref = extremely close)				
Not very close	−4.39	−5.37	−4.38	−4.98
Somewhat close	−2.87	−2.32	−0.67	−0.86
Very close	−0.35	0.55	3.06	2.48
Proportion family (ref = >50%)				
No family members	5.45	−1.82	−1.51	−2.43
Less than 50% family members	5.47	−0.89	0.07	−0.19

**Table 2 ijerph-18-12418-t002:** Bivariate correlations for men (upper triangle; N = 6558) and women (lower triangle; N = 9795). Data from SHARE wave 4.

		1	2	3	4	5	6	7	8	9	10	11	12	13	14	15	16
1	Age		**0.20**	**−0.13**	**0.14**	**−0.04**	**−0.01**	**0.03**	**0.01**	**−0.08**	**−0.07**	**0.00**	**0.03**	**0.01**	**0.01**	**−0.13**	**0.15**
2	Low level of education	**0.23**		**−0.53**	**0.03**	**0.16**	**0.13**	**0.00**	**0.08**	**0.02**	**−0.04**	**−0.09**	**0.11**	**−0.10**	**−0.21**	**−0.16**	**0.11**
3	Medium level of education	**−0.14**	**−0.71**		**0.03**	**−0.04**	**0.02**	**0.03**	**−0.03**	**−0.03**	**−0.02**	**−0.01**	**−0.05**	**0.05**	**0.01**	**−0.01**	**−0.03**
4	Health limitations	**0.18**	**0.09**	**−0.02**		**0.10**	**0.11**	**−0.02**	**0.01**	**0.01**	**−0.05**	**−0.13**	**0.06**	**−0.04**	**−0.07**	**−0.10**	**0.18**
5	Make ends meet with great difficulty	−0.02	**0.11**	**−0.04**	**0.13**		**−0.16**	**−0.21**	**0.07**	**0.04**	**−0.04**	**−0.15**	**0.05**	**−0.14**	**−0.14**	**−0.09**	**0.10**
6	Make ends meet with some difficulty	0.01	**0.10**	**−0.01**	**0.11**	**−0.25**		**−0.21**	**0.07**	**0.04**	**−0.04**	**−0.26**	**0.10**	**−0.16**	**−0.19**	**−0.12**	**0.11**
7	Make ends meet fairly easily	0.03	**−0.01**	**0.02**	**−0.05**	**−0.25**	**−0.48**		**−0.04**	**−0.01**	**0.02**	**−0.03**	**0.01**	**0.03**	**−0.04**	**−0.01**	**−0.05**
8	Family resp. often limits activity	**−0.04**	0.04	−0.03	**0.04**	**0.09**	**0.02**	**−0.04**		**0.04**	**−0.04**	−0.05	0.00	**−0.01**	**−0.10**	−0.05	**0.07**
9	Family resp. sometimes limits act.	**−0.12**	0.02	−0.03	**−0.01**	**0.01**	**0.03**	**−0.03**	**−0.13**		**0.08**	−0.03	0.03	**0.00**	**−0.10**	0.03	**0.02**
10	Family resp. rarely limits activity	**−0.10**	0.00	−0.03	**−0.03**	**−0.03**	**0.00**	**0.03**	**−0.14**	**−0.24**		0.08	−0.02	**0.04**	**0.04**	0.08	**−0.04**
11	Country level social expenditures	0.03	**−0.01**	**−0.05**	**−0.14**	**−0.16**	**−0.20**	**0.03**	−0.01	0.00	0.05		**−0.43**	**0.52**	**0.60**	**0.26**	**−0.14**
12	Unmet healthcare needs (% of pop.)	**0.02**	0.02	−0.01	**0.05**	**0.07**	**0.09**	**−0.02**	**−0.01**	**−0.03**	**−0.01**	**−0.49**		**−0.03**	**0.13**	**−0.10**	**0.09**
13	Trust in the system	**0.02**	**−0.03**	**−0.02**	**−0.03**	**−0.11**	**−0.11**	**0.05**	**0.00**	**0.00**	**0.03**	**0.48**	**−0.02**		**0.37**	**0.16**	**0.01**
14	Trust in other people	**0.06**	**−0.24**	**0.11**	**−0.08**	**−0.14**	**−0.16**	**0.03**	**−0.06**	**−0.10**	**0.00**	**0.49**	**0.17**	**0.34**		**0.20**	**−0.12**
15	Social activities	**−0.17**	**−0.16**	**−0.01**	**−0.17**	**−0.08**	**−0.10**	**0.00**	**−0.01**	**0.00**	**0.07**	**0.24**	**−0.09**	**0.13**	**0.19**		**−0.12**
16	Loneliness	**0.18**	**0.17**	**−0.09**	**0.19**	**0.13**	**0.10**	**−0.05**	**0.09**	**0.02**	**−0.04**	**−0.13**	**0.06**	**0.00**	**−0.16**	**−0.13**	

Note: Correlations cannot be calculated for the two dependent nominal variables (intimate relations, network type). Significant correlations (*p* < 0.05) are shown in bold.

**Table 3 ijerph-18-12418-t003:** Descriptive statistics of ESR by gender, country, and welfare regime. Data from SHARE wave 6.

	Scandinavian				Central Europe									
	Denmark		Sweden		Austria		Belgium		France		Germany		Switzerland	
	M	F	M	F	M	F	M	F	M	F	M	F	M	F
N	480	572	541	685	905	1238	987	1236	893	1241	386	388	810	916
Intimate relations (%)														
Partner, no children	71.0	55.0	79.4	62.6	75.3	44.8	70.3	50.7	72.5	45.6	77.6	65.2	73.3	53.9
Partner and children	0.6	0.2	0.8	0.3	1.2	0.8	1.5	0.9	0.8	0.5	0.8	0.5	2.0	0.9
No partner or children	28.1	44.8	19.3	37.1	23.1	54.2	27.8	48.2	26.7	53.3	21.6	34.4	24.2	45.1
Social network (%)														
Type I (very small, no family)	16.5	7.7	25.7	9.1	21.2	11.5	24.8	11.5	21.2	9.8	18.1	10.8	22.5	10.4
Type II (small, freq. contact)	34.6	39.3	31.6	37.1	42.1	47.5	31.1	35.6	34.6	40.7	36.5	38.1	29.9	36.8
Type III (medium, close contacts)	29.6	31.3	33.8	33.0	18.2	14.5	25.1	27.3	31.5	31.4	29.8	30.7	32.7	31.9
Type IV (large, close contacts)	9.4	21.7	8.9	20.9	17.5	26.5	18.9	25.6	12.8	18.0	15.5	20.4	14.9	21.0
Social participation														
No. of activities/week	0.7	0.8	0.5	0.5	0.3	0.3	0.5	0.4	0.4	0.4	0.4	0.5	0.5	0.4
Lonely (mean)	4.6	4.6	5.0	5.3	4.6	5.0	5.1	5.5	5.0	5.6	4.8	5.1	4.6	4.9
Lonely (%)	8.1	7.1	13.8	22.0	7.0	13.8	16.2	23.2	16.9	26.1	10.0	11.9	8.7	13.2
	**Southern Europe**						**Eastern Europe**							
	**Spain**		**Italy**		**Portugal**		**Czech**		**Estonia**		**Poland**		**Slovenia**	
	**M**	**F**	**M**	**F**	**M**	**F**	**M**	**F**	**M**	**F**	**M**	**F**	**M**	**F**
**N**	**861**	**1066**	**826**	**953**	**433**	**536**	**1032**	**1542**	**1254**	**2136**	**423**	**540**	**502**	**715**
Intimate relations (%)														
Partner, no children	78.1	57.3	81.8	62.4	79.4	60.4	76.0	44.3	72.4	39.6	74.9	49.3	77.4	49.9
Partner and children	4.1	1.7	4.1	3.6	3.0	2.4	1.8	1.3	1.9	0.8	3.9	5.2	5.6	3.6
No partner or children	17.3	40.2	13.8	33.8	17.3	36.6	22.1	54.2	25.2	59.3	21.0	44.3	17.0	45.7
Social network (%)														
Type I (very small, no family)	25.8	13.5	35.1	18.5	27.3	20.7	32.3	17.5	36.4	13.0	38.5	18.0	38.0	16.5
Type II (small, freq. contact)	53.4	60.3	48.8	62.4	52.4	58.8	45.3	53.8	38.1	54.5	41.4	60.9	45.2	60.6
Type III (medium, close contacts)	8.6	8.8	8.2	8.4	10.4	5.2	14.6	14.8	19.8	19.8	15.1	11.1	9.6	9.5
Type IV (large, close contacts)	12.2	17.4	7.9	10.7	9.9	15.3	7.8	13.9	5.7	12.7	5.0	10.0	7.2	13.4
Social participation														
No. of activities/week	0.1	0.1	0.2	0.1	0.2	0.2	0.2	0.2	0.1	0.2	0.0	0.0	0.2	0.3
Lonely (mean)	4.9	5.6	5.5	6.3	5.2	6.0	5.4	5.8	5.3	5.5	5.5	5.9	4.9	5.3
Lonely (%)	15.4	28.8	24.8	40.6	18.7	35.1	21.5	32.5	20.8	26.4	26.3	30.1	12.3	20.1

**Table 4 ijerph-18-12418-t004:** Structural equation modeling with linear and logistic regressions of ESR (wave 6) on micro- and macro-level predictors (wave 4). Men (N = 6558) and women (N = 9795). Odds ratios (intimate relations and network type) or Bs (social participation and loneliness).

	Intimate Relations (ref = No Partner, No Children)				Network Type (ref = Type IV (Largest Network))						Social Part. (No. of Act. ≥ Weekly)		Loneliness	
	Partner, No Children		Partner and Children		Type I (Very Small)		Type II (Small)		Type III (Medium)					
	M	F	M	F	M	F	M	F	M	F	M	F	M	F
Age	0.98 **	0.92 **	0.93 **	0.85 **	0.99	1.03 **	1.00	1.08 **	0.99	1.00	−0.01 **	−0.01 **	0.03 **	0.04 **
Education (ref = high)														
Low	0.72 **	1.21 *	0.89	2.94	1.59 *	1.74 **	1.59 *	1.75 **	0.81	0.81 *	−0.25 **	−0.30 **	0.27 **	0.32 **
Medium	0.87 *	1.08	1.27	2.12	1.38 *	1.34 *	1.38 *	1.50 **	0.94	1.03	−0.14 **	−0.22 **	0.11 *	0.03
Health limitations (ref = no)	0.85 **	0.98	0.73 *	1.10	1.00	1.08	0.97	0.85 **	1.07	1.04	−0.05 **	−0.10 **	0.47 **	0.47 **
Financial situation (ref = easily)														
Great difficulty	0.58 **	0.34 **	1.55	0.57 *	1.00	1.31	1.02	1.33 *	0.77	0.97	−0.11 **	−0.10 **	0.54 **	0.77 **
Some difficulty	0.82 *	0.55 **	2.29 *	0.95	1.12	1.20	1.09	1.10	1.09	0.85	−0.08 **	−0.09 **	0.31 **	0.43 **
Fairly easily	1.04	0.75 **	2.00 *	0.98	0.82 *	1.07	0.90	1.01	0.92	0.97	−0.05 *	−0.07 **	−0.02	0.14 *
Family responsibilities (ref = never)														
Often	1.77 **	1.45 **	1.79	3.33 *	1.05	0.97	1.27	1.06	1.05	0.93	−0.04	0.02	0.39 **	0.59 **
Sometimes	1.87 **	1.88 **	2.21 *	2.70 **	1.14	1.05	1.02	1.06	0.95	0.90	0.07 **	0.01	0.08	0.20 **
Rarely	1.72 **	1.52 **	1.51	1.18	1.02	1.03	0.97	0.97	1.02	0.95	0.07 **	0.07 **	0.00	0.03
Social expenditures (1000 EUR/capita)	1.07 **	1.06 **	1.05	1.07	0.95	1.00	0.97	0.96	1.00	1.01	0.04 **	0.04 **	−0.04 *	0.00
Unmet healthcare needs (% of pop.)	1.04 **	1.05 **	1.09 *	1.11 **	1.08 **	1.01	1.05 *	1.03 *	1.12 **	1.04 *	0.00	0.00	0.01	0.02 *
Individualism (0–100)	0.75 *	0.75 *	0.79	0.65	0.94	0.86	0.75	0.80	1.22	0.87	−0.17 **	−0.23 **	−0.08	−0.18 *
Trust in other people (0–10)	0.97	1.02	0.79	0.69	1.16	0.83	1.28	1.01	0.97	1.19	0.13 **	0.20 **	−0.11 *	−0.22 *
Trust in the system (0–10)	0.99 **	1.00 *	0.98 **	0.99	1.00	1.00	0.99	1.00	1.01	1.02 **	0.00 *	0.00 **	0.02 **	0.01 **

* *p* < 0.05, ** *p* < 0.01. Adjusted BIC for men is 90,049,531 and for women 131,937,863.

## Data Availability

The data are available through www.share-project.org; accessed on 2 September 2020.

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
