# Peer review of "Exclusion from Social Relations in Later Life: Micro- and Macro-Level Patterns and Correlations in a European Perspective"

_ijerph, 2021, doi:10.3390/ijerph182312418_

Round 1

Reviewer 1 Report

The study examines ESR among twelve/fourteen!? European countries at both micro and macro levels by using two datasets. The study reveals some interesting and predictable results. I have some comments:

Specific comments:

            The abstract mentioned 12 countries were examined; however, it is mentioned 14 in the further sections?

Introduction: Pg 2 second paragraph (However xxx..) seems that the research is proceeded by young generation; however, that is not the scope. It needs minor refining.

Pg 3: Why did the author(s) use 60+ ages? What is the definition of the older population? Even if it is 60+ ages, it needs some reference here to justify the selection criteria.

Did the data include more variables or just four categories for micro-level? If there were more than four categories, what was the reason for selecting these 4?

The same comment applies to the macro-level variables, as well…

Data & Methods:

I think the author(s) should revisit the SEM. SEM should be clearly verified on why this method was selected. I feel some hierarchical analyses might be a better fit considering the micro and macro-level of variables.

Discussion: This study did not examine cultural differentiates, and I do not think it is relative to mention.

What are the key findings/results of this study? What is not align with other studies? I think the author(s) should polish this aspect in this section besides predictable gender differences.  

Discussion should touch on some key actions that can be done by local cities as well as non-profit organizations to solve or minimize ESR concerns.

Conclusion: Considering this study's study areas that tend to have much older populations in the near future, the study may play an important role in calling actions.

Author Response

Reviewer #1

  1. The abstract mentioned 12 countries were examined; however, it is mentioned 14 in the further sections?

Thanks for the notice. Now corrected.

  1. Introduction: Pg 2 second paragraph (However xxx..) seems that the research is proceeded by young generation; however, that is not the scope. It needs minor refining.

We have carefully checked this paragraph but could not find a problem regarding inconsistency with the focus on older adults. As this concern was not shared by other reviews, we let the text stand.

  1. Pg 3: Why did the author(s) use 60+ ages? What is the definition of the older population? Even if it is 60+ ages, it needs some reference here to justify the selection criteria.

This is a point well taken. We now explain that we focus on 60+ because it marks the group conventionally (among lay persons and researchers) defined as “older”. See p. 7.

In the literature, papers on “older adults” usually focus on 60+ or 65+, some down to 50+ or up to 70+. The cutoff tends to be rather arbitrary. One advantage with a somewhat lower cutoff (say, 60+ vs. 65+) is that the sample is less selective of the more advantaged (health, psychosocial resources, socioeconomic resources, etc.), especially among lower social strata.

  1. Did the data include more variables or just four categories for micro-level? If there were more than four categories, what was the reason for selecting these 4? The same comment applies to the macro-level variables, as well…

We assume that the reviewer asks about our reasoning for the selection of the micro-level variables (there are six in total) and the macro-level variables. Thank you for the relevant question. Note that we do not claim to be exhaustive in the selection of micro and macro-level variables, and while there may be more micro and macro level variables available, we decided to focus on the most prevalent and/or relevant risk factors for ESR. As written under 2.3, this selection was based on a previous study by Burholt and colleagues (2019).

If, however, the reviews asks about the selection of four indicators of ESR, this explanation is now improved (see comment 3 of reviewer #3 below).

  1. Data & Methods: I think the author(s) should revisit the SEM. SEM should be clearly verified on why this method was selected. I feel some hierarchical analyses might be a better fit considering the micro and macro-level of variables.

The issue of the hierarchy in the data (i.e., nested data within countries) was also mentioned by the editor. This is indeed a good point, but since we have only 12 level-2 units, we decided not to conduct a multilevel analysis as for a multilevel analyses 50 or more level-2 units are necessary to accurately estimate standard errors (Maas & Hox, 2005).

  1. Discussion: This study did not examine cultural differentiates, and I do not think it is relative to mention.

On the one hand we agree, since indeed the study does not include cultural factors. On the other hand, however, we believe that when the models are only able to explain part of the variation, it is relevant (and of interest to many readers) to speculate about omitted factors that could explain the non-explained variance. Cultural factors are obvious candidates, and some of the other reviews have actually suggested to expand this discussion. Hence, we have let the speculation about cultural factors stand.

  1. What are the key findings/results of this study? What is not align with other studies? I think the author(s) should polish this aspect in this section besides predictable gender differences.  

In the Discussion, key findings are first introduced in paragraph 1 (“Findings reveal…”), and then with a general note or consistency with some prior studies. After this, key findings are recognized in the “topic sentence” (first sentence) of each or the following paragraphs (“Our findings align with prior research suggesting that…”, “The loneliness divide can..”, “We also find that women, compared with men, are..”, “A particularly interesting finding is the..”). However, we have now made smaller changes in the discussion to even highlight clearer the key findings and the (in)consistencies with prior findings (especially regarding gender differences that contrasts with some of the literature).

  1. Discussion should touch on some key actions that can be done by local cities as well as non-profit organizations to solve or minimize ESR concerns.

We now discuss at some length some practical implications of our findings. We discuss implications for policy makers and community organizations, with a view to population aging (ref point 9 below).

  1. Conclusion: Considering this study's study areas that tend to have much older populations in the near future, the study may play an important role in calling actions.

See reply above.

Reviewer 2 Report

Paper is well written.

Article considers important topics of ageing societies - the risks of social exclusion for older adults. Paper is well written. What I'm missing is literature review. The chapter with extensive literature review on the topic should be added.

Author Response

Paper is well written.

Article considers important topics of ageing societies - the risks of social exclusion for older adults. Paper is well written. What I'm missing is literature review. The chapter with extensive literature review on the topic should be added.

Reply: The paper already is quite front-heavy due in part to a quite comprehensive review of prior work. However, we agree that some more empirical evidence could be included, and have included a few new references to prior work, e.g. regarding ESR in later life, gender differences, and trust levels across countries.

Reviewer 3 Report

The manuscript "Exclusion from social relations in later life: Micro and macro level patterns and correlates in a European perspective" examines the prevalence and micro and macro level predictors of exclusion of social relations among older adults. It presents a comprehensive perspective on an important topic, especially in relation to examining macro factors. The manuscript is well-written. These strengths noted, some suggestions are provided below.

  1. In page 2 it is claimed that women in Eastern Europe may face higher risks of ESR because of their aging without a partner and their health and financial problems. However, the findings cited in this regard indicate that women report more loneliness but less ESR in relation to more "objective" indicators (such as contact frequency), indicating that perhaps the reason for their greater loneliness does not lie in "objective" factors such as the existence of a partner, but perhaps more "subjective" indicators. The authors might consider making some changes to their argument.
  2. Similarly, the authors might want to cite in the introduction the explanation the loneliness is more pervasive in Southern European countries because of the higher expectation in these countries for closer ties, which is not necessarily met by the families of older individuals.
  3. The introduction could benefit from specifying more clearly which indicators of ESR were chosen and providing a rationale as to why they were chosen.
  4. The variable of family responsibilities is based on an indicator regarding the extent to which the participants feels that such responsibilities prevent them from doing what they want. However, this variable can also be seen as assessing attitudes towards responsibilities and not the extent of care provided. Participants in SHARE are asked about providing care to others, and this variable might more directly measure such responsibilities. It could be useful to explain why the more subjective measure was chosen.
  5. The explanation about macro-level indicators could benefit from more detail about how these indicators were determined. For example – where were the "unmet health care needs" questions taken from? How was the country-level mean calculated?
  6. The authors may want to consider introducing the concept of network typologies earlier in the manuscript, since in the current introduction it isn't clear that typologies will be examined.
  7. In addition, it may improve clarity to provide names for the different typologies and not only numbers in the text.
  8. From the data analysis section, it seems that the dependent and independent variables were measured at different time points. It could be beneficial to explain why this was done instead of, for example, measuring all variables at wave 4 and predicting ESR at wave 5 (in addition to including its baseline measure in the model).
  9. The results could provide a more detailed description of the associations of ESR with macro factors. Especially the direction of the associations could be added to the description.
  10. The authors may want to add a discussion of the results regarding the effects of trust and individualism.

Author Response

The manuscript "Exclusion from social relations in later life: Micro and macro level patterns and correlates in a European perspective" examines the prevalence and micro and macro level predictors of exclusion of social relations among older adults. It presents a comprehensive perspective on an important topic, especially in relation to examining macro factors. The manuscript is well-written. These strengths noted, some suggestions are provided below.

  1. In page 2 it is claimed that women in Eastern Europe may face higher risks of ESR because of their aging without a partner and their health and financial problems. However, the findings cited in this regard indicate that women report more loneliness but less ESR in relation to more "objective" indicators (such as contact frequency), indicating that perhaps the reason for their greater loneliness does not lie in "objective" factors such as the existence of a partner, but perhaps more "subjective" indicators. The authors might consider making some changes to their argument.

Interesting point. We now have broadened this discussion in the paper (see p. 15). It may seem paradoxical that women report more loneliness despite similar levels of contact frequency and social participation. However, women report considerably higher rates of widowhood, and this is likely the main driver behind their relatively high risk of loneliness. That said, more “subjective” factors relating to expectations, needs, standards in the social domain may cause social deficits (i.e. loneliness) among women especially. This is now briefly discussed.

  1. Similarly, the authors might want to cite in the introduction the explanation the loneliness is more pervasive in Southern European countries because of the higher expectation in these countries for closer ties, which is not necessarily met by the families of older individuals.

The paper is already front-heavy, and as these notions are speculations regarding potential unobservable drivers of cross-country differences (see reply to comment by reviewer #1, who actually wanted this discussion dropped) we find it best suited for the Discussion section.

  1. The introduction could benefit from specifying more clearly which indicators of ESR were chosen and providing a rationale as to why they were chosen.

We now provide more discussion and clarification on this issues.

Please note that in the introduction (p. 2) we define ESR:

“ESR is multifaceted in nature and can be understood as a situation of being socially and emotionally disconnected from social relations, with both objective and subjective components to its construction and manifestation (Burholt et al. 2019). While objective ESR can be described as having a limited social network, infrequent social interaction, or lack of participation in social activity, subjective ESR can be characterized by a perceived lack of companionship or feelings of loneliness (ibid.).”

Our choice of indicators aligns with this conceptualization. We now also explain that it aligns with the theoretical notion of “three spheres of sociability” outlined by Gallie et al. (2003).

  1. The variable of family responsibilities is based on an indicator regarding the extent to which the participants feels that such responsibilities prevent them from doing what they want. However, this variable can also be seen as assessing attitudes towards responsibilities and not the extent of care provided. Participants in SHARE are asked about providing care to others, and this variable might more directly measure such responsibilities. It could be useful to explain why the more subjective measure was chosen.

Good point. We now include a brief discussion about this under 2.3 Independent variables. We explain that while caregiving would provide a more direct measure of actual care, the chosen variable in addition highlights whether respondents (in addition to giving care) experience that caregiving compromises other engagements. The latter variable is, in our view, a more direct measure of care as a barrier to social activity/participation/integration.

  1. The explanation about macro-level indicators could benefit from more detail about how these indicators were determined. For example – where were the "unmet health care needs" questions taken from? How was the country-level mean calculated?

Thank you for asking. We have now changed the paragraph to accommodate these concerns.

  1. The authors may want to consider introducing the concept of network typologies earlier in the manuscript, since in the current introduction it isn't clear that typologies will be examined.

Network typologies is used for one (social network) of the four indicators of ESR. We would like to avoid addressing the operationalization of the variables in the Introduction. However, we now add this information at the very beginning of the Dependent variables section.  

  1. In addition, it may improve clarity to provide names for the different typologies and not only numbers in the text.

Agree. This is now done.

  1. From the data analysis section, it seems that the dependent and independent variables were measured at different time points. It could be beneficial to explain why this was done instead of, for example, measuring all variables at wave 4 and predicting ESR at wave 5 (in addition to including its baseline measure in the model).

Thank you for this interesting question. In our reasoning we implicitly assume that our micro and macro level factors are drivers of social exclusion, that is, happening in time before the actual exclusion from social relations. All our micro level factors are measured at SHARE wave 4 and the macro level variables were collected in the years around wave 4, but before wave 6. Information for the outcome variable ESR comes from the first possible follow up measure (which was wave 6). We did not include baseline ESR as we are interested in how micro and macro level factors were associated with the level of ESR not change, which would be the case if we would have included baseline level ESR.

  1. The results could provide a more detailed description of the associations of ESR with macro factors. Especially the direction of the associations could be added to the description.

This is now improved.

  1. The authors may want to add a discussion of the results regarding the effects of trust and individualism.

This is now done.

Round 2

Reviewer 2 Report

The authors have improved the article as requested. Article is suitable for publication in journal.

Reviewer 3 Report

The authors have adequately addressed my comments.